# Low-Temperature Hydrothermal Treatment Surface Functionalization of the Ultrafine-Grained TiMo Alloys for Medical Applications

**DOI:** 10.3390/ma13245763

**Published:** 2020-12-17

**Authors:** Daria Piechowiak, Andrzej Miklaszewski, Mieczysław Jurczyk

**Affiliations:** Institute of Materials Science and Engineering, Poznan University of Technology, Pl. M. Sklodowskiej-Curie 5, 60-965 Poznan, Poland; daria.a.piechowiak@doctorate.put.poznan.pl (D.P.); mieczyslaw.jurczyk@put.poznan.pl (M.J.)

**Keywords:** ultrafine-grained structure, titanium beta alloys, mechanical alloying, low-temperature hydrothermal treatment, alkaline treatment, wettability tests

## Abstract

Hydroxyapatite (HAp) is the most widely used material for bio coating. The functional layer can be produced by many methods, however, the most perspective by its utility, easy to scale up, and simplicity aspects remains a hydrothermal treatment approach. In this work, an HAp coating was produced by low-temperature hydrothermal treatment on the ultrafine-grain beta Ti-xMo (x = 23, 27, 35 wt.%) alloys. The proposed surface treatment procedure combines acid etching, alkaline treatment (AT), and finally hydrothermal treatment (HT). The uniqueness of the approach relies on the recognition of the influence of the molar concentration of NaOH (5 M, 7 M, 10 M, 12 M) during the alkaline treatment on the growth of hydroxyapatite crystals. Obtained and modified specimens were examined structurally and microstructurally at every stage of the process. The results show that the layer after AT consist of titanium oxide and phases based on sodium with various phase relations dependent on NaOH concentration and base composition. The AT in 7 M and 10 M enables to obtain the HAp layer, which can be characterized as the most developed in terms of thickness and porosity. Finally, selected coated samples were investigated in terms of surface wettability test managed in time relation, which for the results confirm high hydrophilicity of the surfaces. Conducted research shows that the low-temperature hydrothermal processing could be considered for a possible adaptation in the drug encapsulation and delivery systems.

## 1. Introduction

Titanium and its alloys are the most commonly used metallic materials in the biomedical field. These biometals have found application in orthopedics, not only maxillofacial but also, dental implants under load-bearing conditions [1]. Their use is justified by low Young’s modulus, low density, and high corrosion resistant properties [1,2].

Recently, researchers have demonstrated that titanium alloys with beta-type structure have become a more enhanced class of biometals [3]. Modern beta type alloys contain nontoxic elements and are characterized by high strength, very low elastic modulus, good corrosion resistance, and excellent biocompatibility in comparison to other biometals [4,5]. An examples of modern beta titanium alloys are Ti-5Mo-5Ag [6], Ti-xMo (x–23%–35wt.%) [5,7], Ti14Zr16Nb [4], Ti23Zr25Nb [4], Ti-6Al-2Nb-2Ta-1Mo [8], Ti-15Mo-Zr [9]. It is worth mentioning that a great effort was put into the research and development of Ti-Mo alloys because of their surgical application [10,11].

Contemporary researchers have demonstrated that the properties of Ti-Mo alloys can be adjusted depending on the Mo content. As the Mo concentration increases, the hardness increases, and the Young modulus decreases [12]. This approach gives the possibility to adapt the properties to the specific application. The addition of approximately 10% Mo, or more, allows only to Ti(β) phase to appear in Ti-Mo alloy, however, a high cooling rate for lower stabilizing element addition [7] remains needed. Additionally, the properties of alloys do not depend only on chemical composition, but also on microstructure modification [3]. They can be enhanced by nano-structurization. To receive the nano- or ultrafine-grained structure, the mechanical alloying (MA) process can be applied. New prospects appear with nanostructure materials, which exhibit better mechanical and physicochemical properties in comparison to their microcrystalline counterparts. Recent studies have shown that nanostructuring of titanium can considerably improve not only the mechanical properties but also the biocompatibility [13].

In general, biometals possess lower biocompatibility and osseointegration characteristics in comparison to bioceramic materials. To improve the above, a bioceramic covering can be applied [14]. The most common approach in commercial practice uses the hydroxyapatite (HAp) covering [15,16].

Hydroxyapatite is one of the largest groups of calcium phosphate-based materials. The chemical composition of HAp is [Ca_10_(PO_4_)_6_(OH)_2_]. HAp possesses chemical similarity to the mineral component of bones and hard tissues [15]. Moreover, HAp can be characterized by high biocompatibility, bioactivity, and resorbability. In terms of thermodynamics, it also provides the highest stability in body fluids among all calcium phosphate salts [17]. Due to its high bio properties, HAp has found a wide range of applications—it is used as materials for implants and prostheses; as a coating material to improve biocompatibility and osteointegration; as also drug delivery systems to control the release of drugs [18].

HAp coating can be produced by a plasma spraying [19], immersion in simulated body fluid [20], laser deposition [21], ion beam dynamic mixing [22], ion beam deposition [23], sol-gel precipitation [24], hydrothermal treatment [25,26], ion implantation, alkaline treatment [27], electrochemical methods [24], and others. Plasma spraying is the most common commercial method used to cover titanium implants [28]. However, this technique may not provide an equal thickness especially in the complex geometries, sufficient adhesion to the substrate, and characterize the HAp-cover by inhomogeneous phase composition and low crystallinity [29].

The alternative approach, without the disadvantages like plasma spraying, is for example the hydrothermal treatment [28]. This method enables the creation of a high-quality HAp-cover with Ca/P ratio similar to the stoichiometric value of HAp. Furthermore, it is possible to control the thickness of the layer or morphology, size, and orientation of HAp-grains [25].

To embed HAp cover on the titanium substrate, there is a need for specific surface activation. The most frequently applied methods are alkali treatment [28], acidic treatment [25], and micro-arc oxidation (MAO) [30]. In this work, the alkali treatment approach is carried out, focusing the conducted research on its simplicity understand as easy to control, repeatable, efficient and, also, easy to scaleup process. The most common in use solution for the alkali treatment is 5 M NaOH [27,28,31]. The purpose of the alkaline treatment is to create the oxide-hydroxide layer on the titanium surface with the specific topography and positive surface charge [32]. As a result of the alkaline treatment, the surface bioactivity is enhanced. It enables the further formation and strong bonding of the HAp coating towards the original surface [33,34]. Such a process can be observed under the condition of a positively charged layer attracting the negatively charged phosphates ions, which allows the absorption of calcium ions on the treated surface [32].

Hydrothermal treatment is based on the chemical dissolution of HAp crystals from an aqueous solution with calcium and phosphate precursors. This process is carried out in an autoclave or pressure vessel at elevated temperature and pressure [32]. During the hydrothermal treatment, the pH and temperature control of the solution is crucial [35]. The HAp crystals morphology and, also, adhesion between the embedded surface and HAp coating strongly depends on pH and temperature factors [36,37,38]. Moreover, the temperature is related to the phase purity and Ca/P ratio. To begin the chemical deposition the boiling point of the solution has to be achieved, although the higher temperature the results are better. Regarding the relation between pH and HAp crystals morphology, the high pH value facilitates the growth of a spherical or short rod-like form of HAp [16]. However, rod- or plate-like structures appear under acidic or neutral conditions. Furthermore, hydrothermal treatment can differ in the chemical composition of the solution. Calcium chelating agents and various organic surfactants are the most common HAp sources for hydrothermal treatment. Although, EDTA, as the most used calcium chelating agent, found the application in HAp production. Ca-EDTA is simple to decompose under hydrothermal conditions. This modifier enables the formation of longer HAp crystals at a lower temperature.

The present study investigates the formation of the hydroxyapatite layers on the TixMo (x = 23, 27, 35 wt.%) ultrafine-grained alloys obtained by mechanical alloying and hot pressing approach. The HAp covers were obtained after the proposed low-temperature procedures. Surface treatments rely on acidic etching, alkaline, and finally hydrothermal treatment. All process stadiums were investigated structurally and was depictured on the schematic process illustration in Figure 1.

A low-temperature surface treatment processing applied for sintered ultrafine-grained TixMo alloys remain attractive for further recognition in drug encapsulation and delivery systems. Processing temperatures and applied for treatment solutions for the above stay relevant factors.

The starting sample compositions and in the same resulting phase relation with its inherent structure and size, influence as the research shows a step-by-step processing relation. The phase grain boundaries and the primary powder particle grains that could be noticed for the sinters obtained in a specific way (revealed after etching), by its energetically favorable state influence further processing stadiums that mimic the substrate structure. Analyzed in this work, various molar concentrations of NaOH (5 M, 7 M, 10 M, 12 M) alkaline solutions shows its influence on the treatment products, necessary however for further proper HAp layer formation and growth. The above points out also the dependence between the substrate activity characteristics and the applied solution concentration. Analyzed additional wettability characteristics of the obtained HAp layers correlate with not modified surfaces and base microcrystalline titanium, confirming its meaning and usability in a broad medical applications range.

## 2. Materials and Methods

### 2.1. Materials and Specimens

For the hydrothermal treatment surface functionalization, an ultrafine-grain base Ti23Mo, Ti27Mo, and Ti35Mo (wt.%) alloy samples were fabricated by mechanical alloying (MA) and powder metallurgy (PM) methods, which for the details of preparation could be found in our earlier work [5,7]. The sintering procedure proposed in this work was the same for all samples and relay on the hot pressing in 800 °C in vacuum conditions which was schematically shown in Figure 2. To prepare the (β) titanium alloys, the commercial Ti (Alfa Aesar, 99.9% purity, CAS:7440-32-6, Karlsruhe, Germany) and Mo (44 μm, 99.6%, CAS:7430-08-7 Sigma Aldrich, Karlsruhe, Germany) powders were used. The dimensions of the hot-pressed cylinder shape bulk samples were 5 mm high and 8 mm in diameter. In the next step, samples were mechanically abraded with 180 to 2500-grit silicon carbide (SiC) grinding papers followed by polishing with 0.5 μm diamond paste and cleaning in ethanol. Prepared samples were chemically etched in an acid solution of 3% HF + 10% H_3_PO_4_ (Poch S.A., Gliwice, Poland) for 3 min and afterward washed in distilled water.

### 2.2. Alkaline Treatment

Alkaline treatment (AT) of TixMo (x = 23, 27, 35) alloys were performed in four different molar concentrations of NaOH (Poch S.A., Gliwice, Poland). Each alloy set of Ti23Mo, Ti27Mo, and Ti35Mo samples was treated in 5 M, 7 M, 10 M, and 12 M NaOH solution. The samples were anchored to the bottom of the glass beakers and filled with the 250 mL of NaOH solution. Subsequently, the beakers were put into the furnace and treated for 24 h at 90 °C conditions. At the end of the alkaline treatments, samples were washed in distilled water. After AT, the samples were marked as Table 1 shows.

### 2.3. Hydrothermal Treatment

Hydroxyapatite coatings were synthesized from CA-EDTA solution under hydrothermal conditions. The preparation of the solution was divided into two stages. Firstly, Ca-EDTA (Sigma-Aldrich, CAS: 62-33-9, Saint Louis, MO, USA) and KH_2_PO_4_ (Sigma-Aldrich, CAS: 7778-77-0, Saint Louis, MO, USA) were dissolved in deionized water with 0.25 mol/L and 0.15 mol/L, respectively, to prepare 200 mL of solution. Secondly, the pH of the solution was titrated by 1 mol/L NaOH to the value of 8.9. The fresh resulting mixture was transferred into glass beakers with placed earlier prepared AT samples, that were next sealed and put into the furnace. The hydrothermal treatment of the samples was conducted at 130 °C for 6 h (time was measured from the time when the boiling point was reached) with the next cooling to room temperature with the furnace. The sample was removed from the beakers and dried at room temperature. After hydrothermal treatment (HT), the samples were marked as presented in Table 2.

### 2.4. Materials Characterization

At every stage of sample preparation, an X-ray diffraction (XRD) (Empyrean, Panalytical, Almelo, Netherlands) analysis was conducted by the Panalytical Empyrean equipment with the copper anode (CuKα—1.54 Å) at a Brag-Brentano reflection mode configuration with 45 kV and 40 mA parameters. The measurement parameters were set up for 20°–90° with a 15 s per step 0.0167° in all cases.

The following structural models were used:(1)Base specimens:
Ti(α)—ref. code 04-003-5042Ti(β)—Ti_0.9_Mo_0.1_—ref. code 04-018-6034(2)Alkali treatment:
TiO—ref. code 01-086-2352Na_2_Ti_2_O_4_(OH)_2_—ref. code 00-057-0123Na_0.55_Mo_2_O_4_—ref. code 00-040-1022(3)Hydrothermal treatment:
Ca_10_(PO_4_)_6_(OH)_2_—ref. code 01-080-3958Na_2_HPO_4_—ref. code 00-001-0997

For detailed structure analysis, a collation of the pairs of the highest intensity peak relations of processing product with the substrate was revealed. The data refers to obtained patterns of the processing products depicted in a graph relation, which for a proportion of the highest intensity peak of the substrate were related to the highest intensity peak of the product phases. Indirectly presented data represent a quantitative collation, which for diminished base substrate intensity with increasing intensity of products corresponds to a layer thickness growth. Additionally, the growing intensity of the product phases corresponds to its higher crystallinity and volumetric amount related to the collected instrument data. Further conclusions could also be drawn from the presented data.

Scanning electron microscopy in the SE-mode (SEM, MIRA3 Tescan) was used to characterize the surface after all stages of modification.

The contact angles (CA) of analyzed surfaces were determined by the sessile drop method. The measurements were performed by a Drop Shape Analyzer-DSA25 instrumentation (KRÜSS GmbH, Hamburg, Germany) and the KRÜSS ADVANCE 1.5.1.0 software instrumentation (KRÜSS GmbH, Hamburg, Germany). The used parameters were as follows:measuring liquid: glycerol,drop volume: 0.5 µL,dosing speed 0.2 mL/min,measuring time: 8 s,probing frequency for multiple measurements: 50 fps,base cut off: automatic/manual,CA fitting method: Young Laplace,measurements conditions: ambient 23 °C.

## 3. Results and Discussion

In the present study, the hydroxyapatite cover on the ultrafine-grained beta titanium alloys was obtained and afterward evaluated focusing on its formation criteria. Using the various molar concentrations of NaOH (5 M, 7 M, 10 M, 12 M) the influence of the alkali treatment on HAp layer growth were investigated, while the rest of the parameters were constant. Furthermore, the chemical composition impact of beta titanium ultrafine-grained alloys on the surface modification approach was also investigated.

### 3.1. Base Materials—Ti-Mo Systems

Figure 3 shows the XRD patterns of Ti-xMo bulk sample ultrafine-grained alloys. In all of the analyzed cases, the main phase corresponds to Ti(β) type structure—Ti_0.9_Mo_0.1_. However, for all systems, a small amount of the Ti(α) phase could be noticed. The amount of Ti(α) phase remains strictly connected with the beta stabilizing Mo content in the starting powder mixtures. Additionally, an increase in molybdenum content causes also a decrease in the beta phase unit cell, which is revealed by a peak shift to larger angles. The reductions in the interplanar distance (Δd_hkl_) between references phase and MA alloys set as follows 0.002513, 0.004263, and 0.014648 by increasing molybdenum content in the alloy (Figure 3). The above relation could be only explained by the bcc unit cell transition from titanium-based one to molybdenum which for the cell dimension is lower. In the specific compared relation of Ti_0.9_Mo_0.1_ to Mo lattice parameters dimension present as fallowed 325.7 to 314.7 pm. A similar effect was already observed in our earlier research [5] which details structural estimation confirming such behavior; however, other intermediate beta phases with growing molybdenum content were confirmed like MoTi.

A hot pressing of the prepared by MA blends allows obtaining alloys with an ultra-fine grain structure for all proposed compositions, as shown in Figure 4. The average grain size of Ti23Mo oscillates 1.097 ± 0.216 (Figure 4). The microstructure remains homogenous and shows a polygonal grain morphology suggesting fine kinetics between the components and stable state of the obtained systems.

### 3.2. Acid Etching

The chemical etching step was applied in the proposed procedure to develop a surface roughness. Figure 5 shows the SEM microphotographs of Ti23Mo, Ti27Mo, and Ti35Mo surface after etching in 3% HF + 10% H_3_PO_4_ acid solution. The etching process runs firstly along the primary powder particle grains, some of the effects could be also observed for the phase grain boundaries. In the case of acid etching of Ti27Mo, the obtained effect remains inhomogeneous. Etching takes place on the entire surface along the powder particle grains, but the grain boundaries etching can be indicated in some areas. Ti35Mo alloy is the most resistant to acidic etching in the proposed process solution. The primary powder particles cannot be pointed out. The etching process runs along the grain boundaries, but the final effect remains fine.

### 3.3. Alkali Treatment in NaOH Solution with Different Molar Concentration

Figure 6, Figure 7 and Figure 8 shows the XRD patterns of the alkali-treated surface of the TixMo alloys. The diffraction patterns point out mostly Ti(β) and Ti(α) phases derived from the substrate. However, the titanium oxide and phases base on sodium (Na_2_Ti_2_(OH)_4_ and Na_0.55_Mo_2_O_4_) can be indicated. The peaks from the layer phases are characterized by low intensity. This indicates a slight thickness of the created layers. Ti35Mo sample possesses the highest resistance to alkaline treatment. XRD patterns of Ti35Mo do not reveal Na_2_Ti_2_(OH)_4_ phase regardless of the NaOH concentration. The largest quantitative share of the Na_2_Ti_2_(OH)_4_ phase can be observed for the samples 23-7AT and 27-7AT.

The surface morphology analysis after alkaline treatment with the various NaOH solution for 24 h at 90 °C, was shown in Figure 9. The results indicated that the surface morphology depends on the molar concentration of NaOH solution and, also, the chemical composition of the base materials. In all cases, the created layers are characterized by surface cracking. Formed structure imitates the primary powder and phase grains because of the privileged higher energy regions with easier kinetics. For the abovementioned, it is worth noticing that the reaction kinetics indirectly depend on a substrate relation and solution concentration, which may lead to an unexpected product phase stability diminishment, which, in our case, is visible in an intense layer cracking and delamination from the substrate. The depth of cracks and the size of formed structures have the smallest value in the case of Ti35Mo alloy.

Additionally, NaOH-soaked samples (at all concentrations) exhibit a porous cellular-looking morphology. Only the sample Ti35Mo 10 M possesses a unique microstructure morphology, which can be determined as an island-like morphology.

### 3.4. Hydrothermal Treatment in Ca-EDTA Solution

The XRD patterns of hydrothermally treated TixMo alloys confirm the presence of HAp on the sample surfaces, as shown in Figure 10, Figure 11 and Figure 12. The XRD patterns obtained for the samples after AT in 7 M, 10 M, and HT are characterized by sharp and high-intensity HAp peaks. Additionally, no peaks correspond to any other crystalline calcium phosphate phases. Disodium phosphate (Na_2_HPO_4_) impurities could be indicated in the layer, which derives from the Ca-EDTA solution. This phase forms a binder between the hydroxyapatite crystals. The patterns obtained for the samples after AT in 12 M and HT are characterized by a low intensity of the peaks originating from the HAp phase.

The overall morphology of the obtained after HT process surfaces, indicates the existence of many uniform cavernous-like HAp microstructure with needle-like HAp particles, as is presented in Figure 13. However, preceding alkaline treatment has a significant influence on HAp formation. SEM microphotographs of 23-7HT and 35-7HT show HAp covers on Ti(β) alloys, which are characterized by high surface development and open porosity. HAp layers formed on samples after 10 M AT are consistent and possess pores with a larger dimension than samples after 7 M AT. It is relevant for the biomedical application when surfaces that contact directly with a tissue possess the possibilities to be colonized and overgrow by bone tissue. As the molar concentration of NaOH increased from 10 M to 12 M, the HAp phase does not cover the entire surface and exhibits an island-like morphology. The abovementioned conclusion indicates low stability of the AT products in the case of 12 M NaOH solution, which does not provide a proper anchor for the HT product phase.

Structural collated data in the graph (Figure 14), represents the ratio of the intensity between the peak of the post-products phases and the Ti(β) phase, which shows the differences in the reaction after AT and the next HT. It is worth noticing that the peak ratio relates to the highest intensity peaks for each phase that corresponds to the following family of the plane: (111) for TiO-cubic, (020) for Na_2_Ti_2_O_4_(OH)_2_-orthorhombic, (−102) for Na_0.55_Mo_2_O_4_-monoclinic, (121) for HAp-hexagonal, and also (110) for Ti(β) phase. The abovementioned data confirm the earlier conclusion about the processing products, that mimic the substrate, which was also confronted with the SEM analysis discussion.

Regarding the AT product ratio presents a complex overview. Although, it does not reveal the relation between solution concentration or substrate composition and surface response. Locally, for the same substrate composition, some tendency could be observed. For Ti23Mo example, the TiO and Na_2_Ti_2_O_4_(OH)_2_ product phase amount grows when the solution concentration increases from 5 M to 7 M. However, further 10 M and 12 M treatment exposes a decrease or even absence in the case of TiO product phase. A familiar relation could be observed in the case of the Ti27Mo sample, although only for the Na_2_Ti_2_O_4_(OH)_2_ product phase.

The represented data confirms the HAp structural response after the HT in all analyzed experimental examples. However, in some cases, in a very low degree. In the case of the HT products, it stays important that the related value of I_HAp(121)_/I_β(110)_ ratio depends indirectly on earlier AT treatment. Although, AT products were not noticed in the patterns after HT even when an HAp intensity was very low. The reason for this behavior is probably connected with the low stability of products after AT. The obtained results were assessed in terms of structural relations, understood as the highest HAp intensity ratio. This criterion indicates the samples after 10 M AT.

### 3.5. Wettability Tests

The properties of the surface, including its hydrophilicity or hydrophobicity, have a significant influence on the interaction between the surface of the biomaterial and the biological environment, in particular, on the number of adsorbed proteins and their adhesion to the surface. In biomedical applications, a desirable situation is defined by a material that exhibits hydrophilic surface properties. The abovementioned properties have a positive effect on the adsorption, adhesion, and cell proliferation activity, which remain important in osseointegration characteristics improvement.

Selected HAp layers on TixMo alloys have been characterized in terms of wettability by glycerol testing fluid presented in Figure 15. All modified sample surfaces possess high hydrophilicity. The contact angle (CA) in time when the drop was placed (t = 0) was 30°, 26°, and 31.5° for 23-10HT, 27-10HT, and 35-10HT, respectively. In the case of 23-10HT after 4.5 s the CA decreased to approximately 7.5°, the same effect was achieved after 6.5 s for 27-10HT and 35-10HT. Additionally, the unmodified surfaces of Ti-xMo alloys and microcrystalline titanium were analyzed for comparison purposes. The HT confirms a visible CA decrease for selected covered samples in comparison to untreated samples.

## 4. Conclusions

The authors proposed a schematic course of the process presented in Figure 16. The attention should be focused on the substrate composition, structure, size, morphology, homogeneity, and physiochemical properties that follow, influencing HAp layer state and final properties through the intermediate alkaline treatment, that mimics the substrate.

The obtained in this work results indicate that the proposed process enables the preparation of a hydroxyapatite coating. However, the HAp layer is the most developed for samples treated with alkaline treatment in 7 M and 10 M NaOH solution. The above statement confirms also stable intermediate phase forms obtained on the substrates after AT, necessary as Figure 16 shows for HAp layer formation and growth.

The sample comparison in terms of starting chemical composition does not reveal any significant differences between the resulting coatings for the samples after treatment with 7 M and 10 M NaOH. The differences are noticeable only for the samples after treatment with 12 M NaOH, the least developed coating was obtained for the Ti27Mo 12 M sample. Coatings on Ti23Mo 12 M and Ti35Mo 12 M samples have similar characteristics and island-like morphology. Furthermore, conducted research confirms the positive effect of the obtained HAp layers on the wettability surface characteristics in the time managed tests.

The obtained results prove that a proper process control allows overcoming mentioned problems for other technological approaches of HAp deposition, also in large scale and complex parts geometry at a low demandable approach. The hydrothermal treatment approach remains to grow in interest by the research community, and in the future, we hope by its simplicity, also by the manufacturers. Conducted research remains also the first step in further consideration and field recognition in a possible adaptation of low-temperature hydrothermal processing in the drug encapsulation and delivery systems.

## Figures and Tables

**Figure 1 materials-13-05763-f001:**
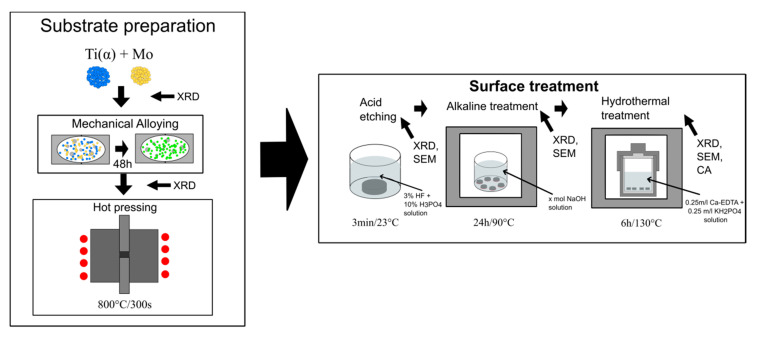
Schematic process illustration.

**Figure 2 materials-13-05763-f002:**
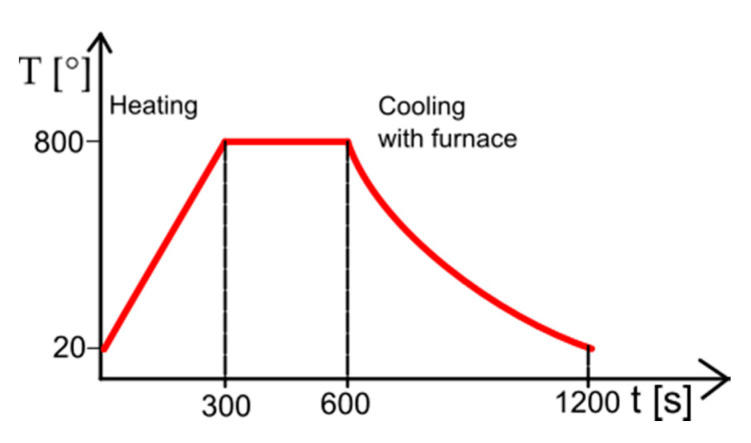
Schematic representation of the hot pressing process.

**Figure 3 materials-13-05763-f003:**
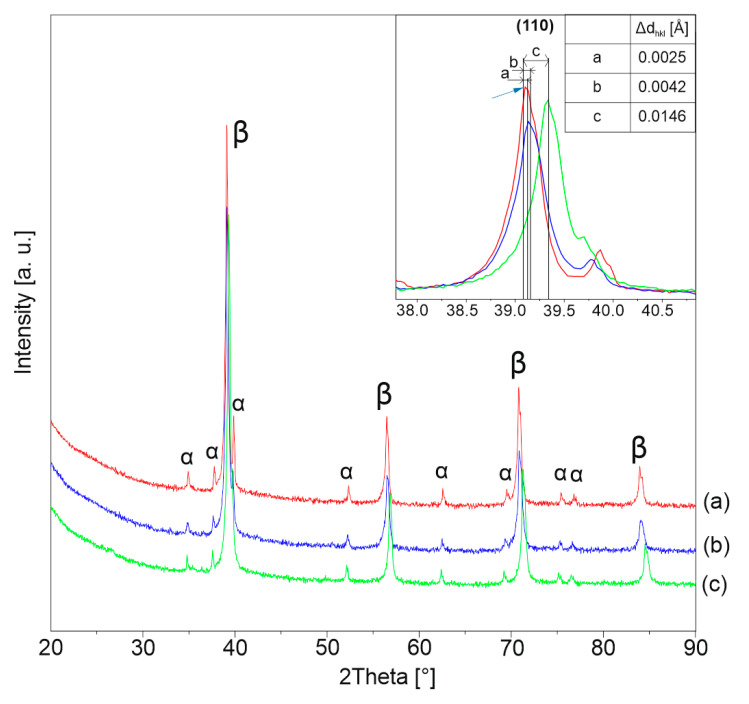
XRD patterns of sintered specimens: (**a**) Ti23Mo, (**b**) Ti27Mo, (**c**) Ti35Mo.

**Figure 4 materials-13-05763-f004:**
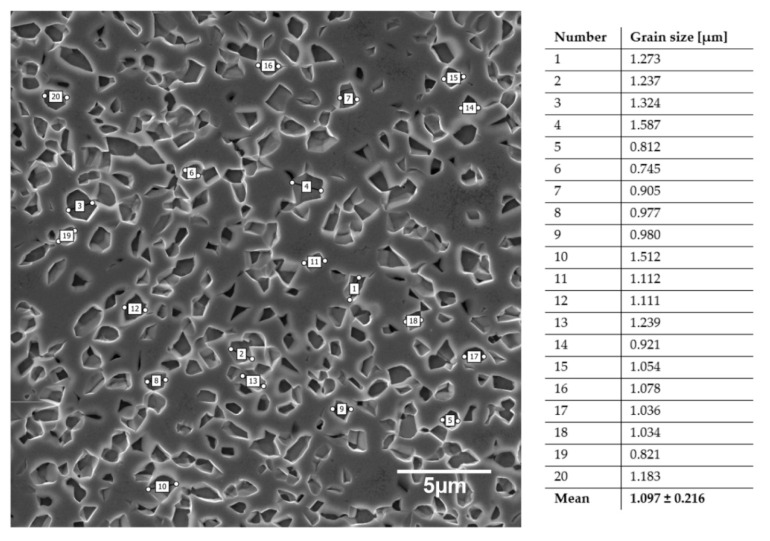
SEM microphotographs of Ti23Mo and the determination of average grain size.

**Figure 5 materials-13-05763-f005:**
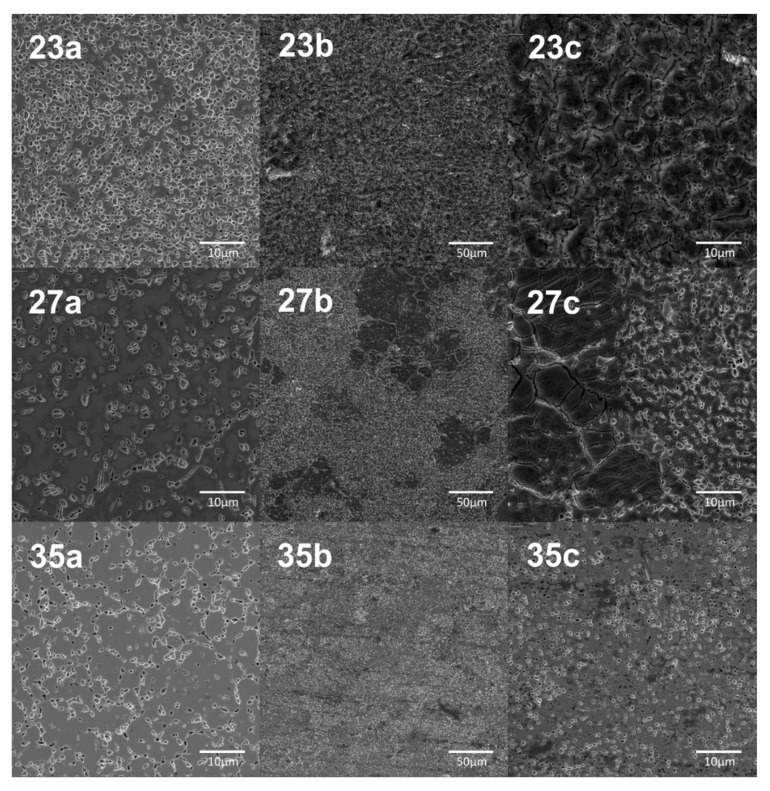
SEM microphotographs of Ti23Mo, Ti27Mo, Ti35Mo base materials—23a, 27a, 35a, respectively, and after acid etching at the different magnification (23b, 23c, 27b, 27c, 35b, 35c).

**Figure 6 materials-13-05763-f006:**
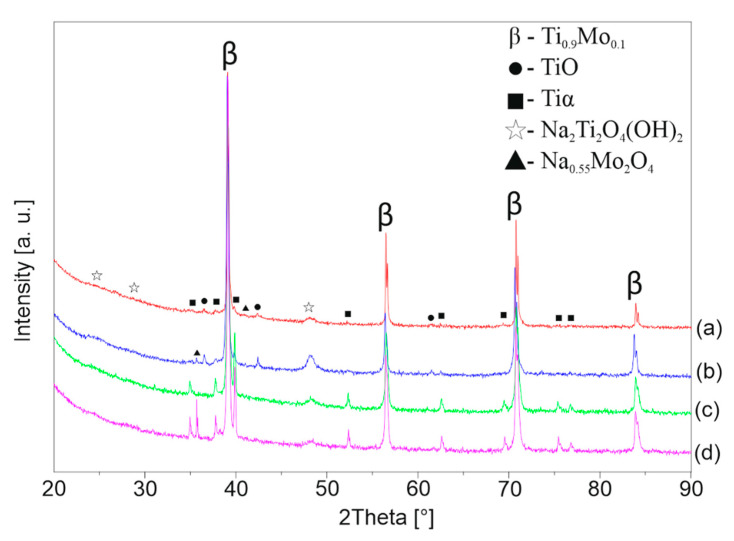
XRD patterns of Ti23Mo after alkaline treatment in (**a**) 5 M, (**b**) 7 M, (**c**) 10 M, (**d**) 12 M NaOH solutions.

**Figure 7 materials-13-05763-f007:**
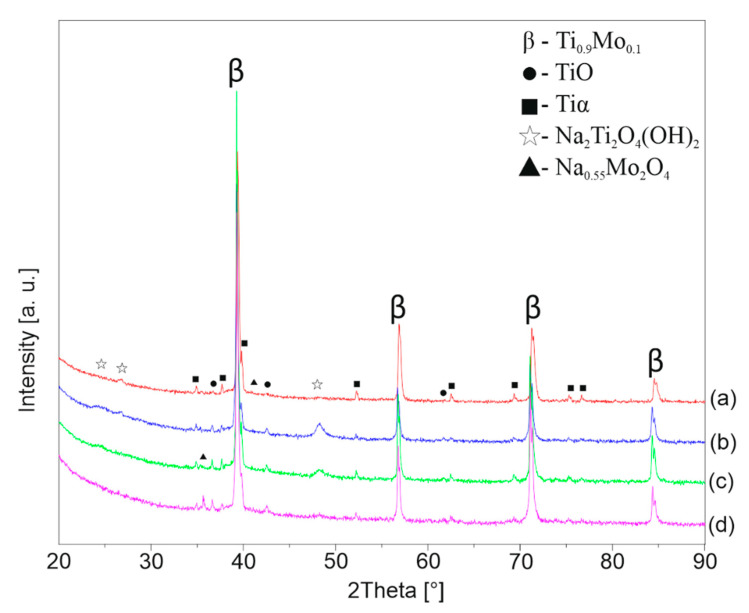
XRD patterns of Ti27Mo after alkaline treatment in (**a**) 5 M, (**b**) 7 M, (**c**) 10 M, (**d**) 12 M NaOH solutions.

**Figure 8 materials-13-05763-f008:**
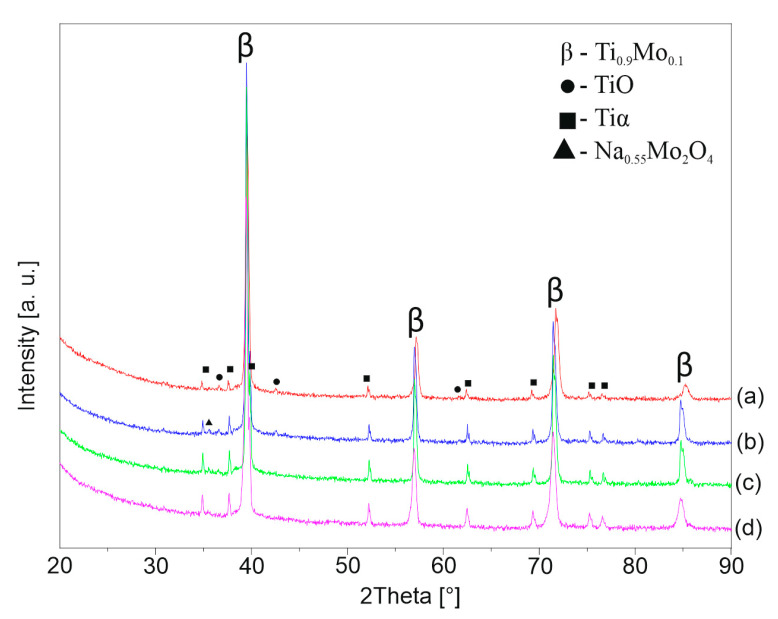
XRD patterns of Ti35Mo after alkaline treatment in (**a**) 5 M, (**b**) 7 M, (**c**) 10 M, (**d**) 12 M NaOH solutions.

**Figure 9 materials-13-05763-f009:**
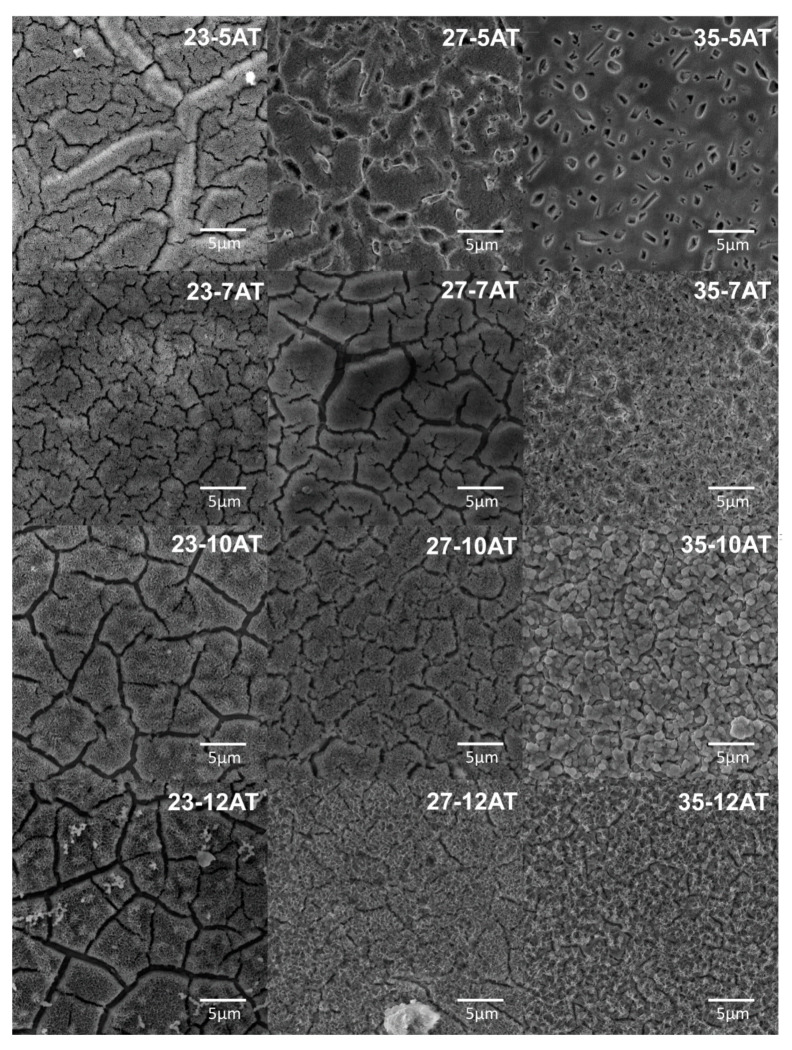
SEM microphotographs of Ti23Mo, Ti27Mo, Ti35Mo, after acid etching and alkaline treatment in various molar concentrations of NaOH.

**Figure 10 materials-13-05763-f010:**
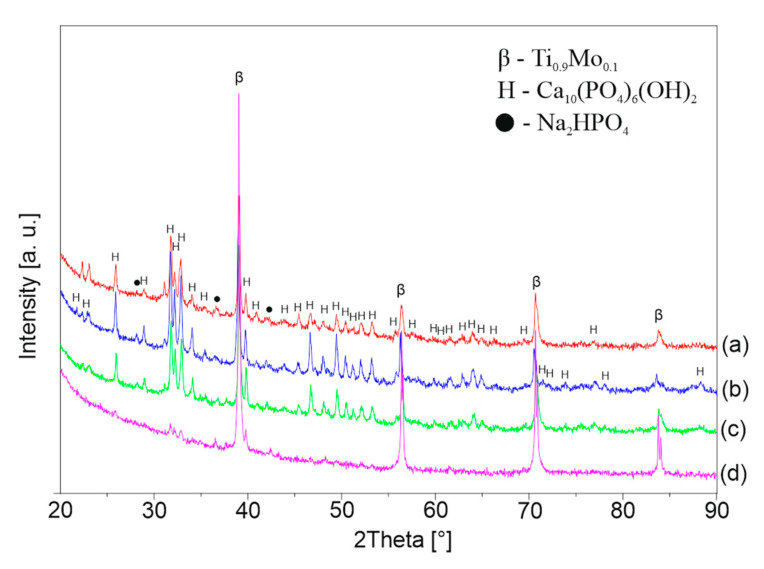
XRD patterns of (**a**) 23-5HT, (**b**) 23-7HT, (**c**) 23-10HT, (**d**) 23-12HT.

**Figure 11 materials-13-05763-f011:**
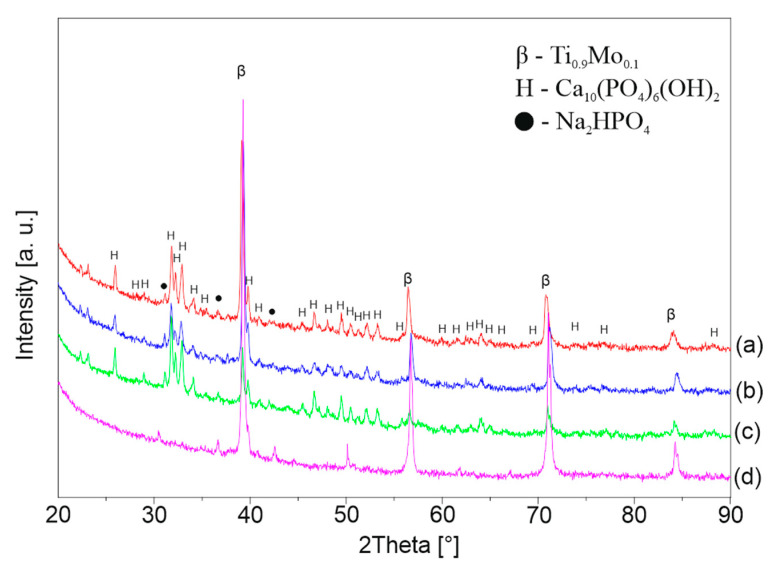
XRD patterns of (**a**) 27-5HT, (**b**) 27-7HT, (**c**) 27-10HT, (**d**) 27-12HT.

**Figure 12 materials-13-05763-f012:**
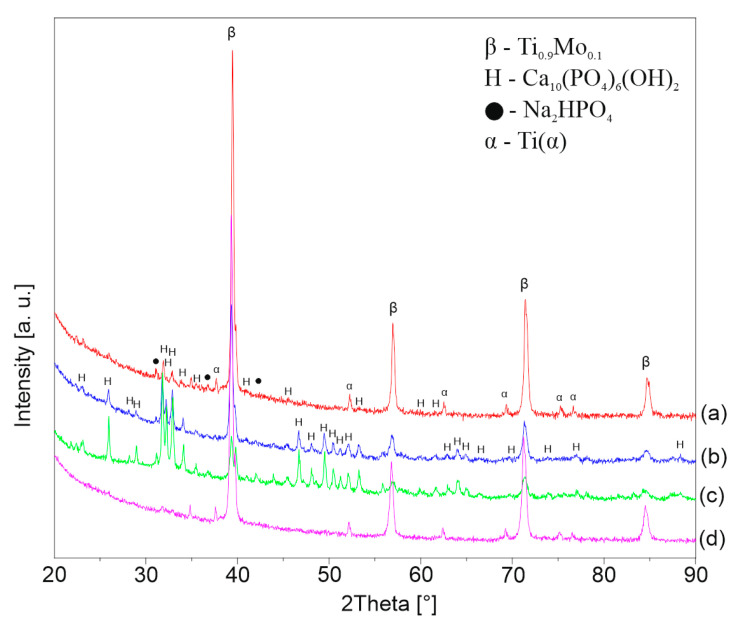
XRD patterns of (**a**) 35-5HT, (**b**) 35-7HT, (**c**) 35-10HT, (**d**) 35-12HT.

**Figure 13 materials-13-05763-f013:**
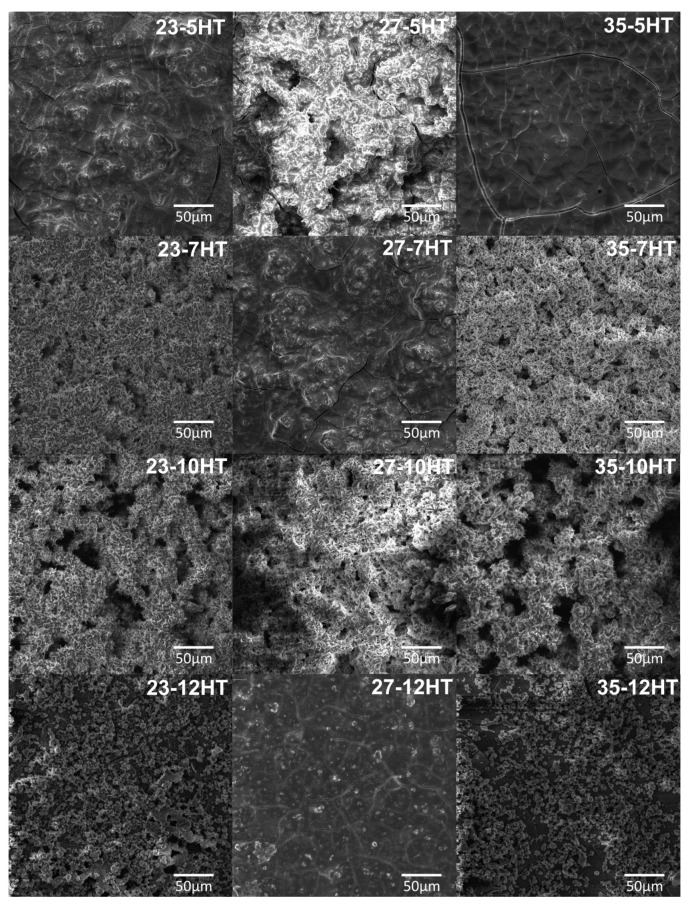
SEM microphotographs of Ti23Mo, Ti27Mo, Ti35Mo, after acid etching, alkaline treatment in various molar concentrations of NaOH, and hydrothermal treatment.

**Figure 14 materials-13-05763-f014:**
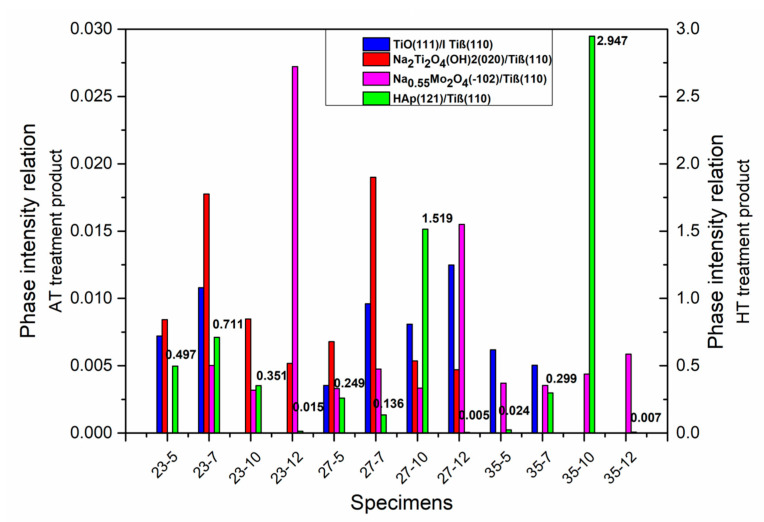
The intensity pairs of the products phase relation values collated in changeable substrate composition and AT treatment solution concentration.

**Figure 15 materials-13-05763-f015:**
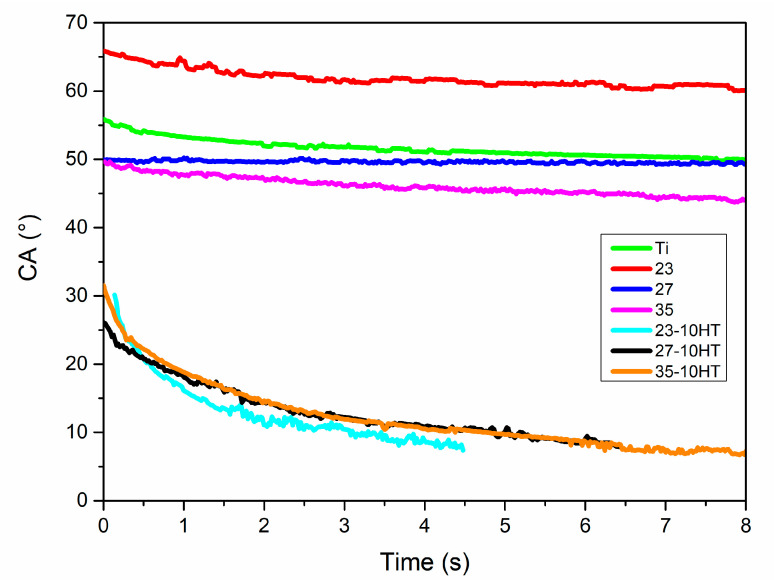
A glycerol contact angle relation for 23-10HT, 27-10HT, 35-10HT, Ti23Mo, Ti27Mo, and Ti35Mo samples in a time function.

**Figure 16 materials-13-05763-f016:**
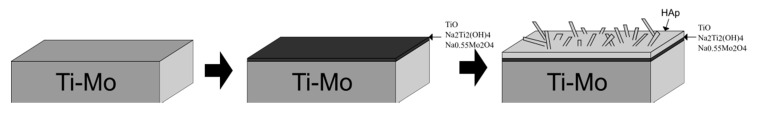
The schematic presentation of HAp formation on TixMo specimens.

**Table 1 materials-13-05763-t001:** Sample indication after alkaline treatment.

Sample/Molar Concentration of NaOH	5 M	7 M	10 M	12 M
Ti23Mo	23-5AT	23-7AT	23-10AT	23-12AT
Ti27Mo	27-5AT	27-7AT	27-10AT	27-12AT
Ti35Mo	35-5AT	35-7AT	35-10AT	35-12AT

**Table 2 materials-13-05763-t002:** Sample indication after hydrothermal treatment.

Sample/Molar Concentration of NaOH	5 M	7 M	10 M	12 M
Ti23Mo	23-5HT	23-7HT	23-10HT	23-12HT
Ti27Mo	27-5HT	27-7HT	27-10HT	27-12HT
Ti35Mo	35-5HT	35-7HT	35-10HT	35-12HT

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
