# Peer review of "Low-Temperature Hydrothermal Treatment Surface Functionalization of the Ultrafine-Grained TiMo Alloys for Medical Applications"

_materials, 2020, doi:10.3390/ma13245763_

Round 1
Reviewer 1 Report
This is a manuscript on the effects of various chemical treatments on Ti-xMo alloys followed by HA coating that was conferred by low-temperature hydrothermal treatment. The characterization was performed by XRD, SEM, contact angle measurement. This manuscript was well written focusing on this specific type of surface modification on Ti-xMo alloys, however, the impact of the provided information would be limited. Authors should dive into deep to examine how those surface modifications could affect the cells and surrounding tissues. I would suggest in vitro test using osteoblast-like cells to compare the biocompatibility of those surfaces.
Authors claim that 7M and 10M provided best effects, however, other previous reports demonstrated alkaline treatments conferred sub-micro surface topographies that led to higher biocompatibility. Authors need to look into higher magnification images to claim which concentration is the best treatment.
Minor issues
Abstract
Authors should be more specific on what was the “best effect”.
Line 58
The term “bio-film” could remind most readers of the film created by bacteria. Authors need to rephrase the term.
Figure 1
Please spell out MA, though it was spelled later in the text.
Figure 4
What is the difference between a and c? They looked like images with the same magnification.
Author Response
Dear Reviewer 1,
We would like to appreciate you allowing us to resubmit our manuscript: Low-temperature hydrothermal treatment surface functionalization of the ultrafine-grained TiMo alloys for medical applications to Materials. We are pleased with your comments and objective feedback considering the draft of our article. The changes were marked in our article in red colors.
All of the suggestions were included during the correction of our article and the comments were appropriately responded:
This is a manuscript on the effects of various chemical treatments on Ti-xMo alloys followed by HA coating that was conferred by low-temperature hydrothermal treatment. The characterization was performed by XRD, SEM, contact angle measurement. This manuscript was well written focusing on this specific type of surface modification on Ti-xMo alloys, however, the impact of the provided information would be limited. Authors should dive into deep to examine how those surface modifications could affect the cells and surrounding tissues. I would suggest in vitro test using osteoblast-like cells to compare the biocompatibility of those surfaces.
Thank you for this comment, however, at the present stage of the studies above analysis outreach planned scenario. The authors consider those research in further steps when the active pharmaceuticals will be introduced into the HAp layer.
Authors claim that 7M and 10M provided best effects, however, other previous reports demonstrated alkaline treatments conferred sub-micro surface topographies that led to higher biocompatibility. Authors need to look into higher magnification images to claim which concentration is the best treatment.
Thank you for this comment. The authors concluded from the obtained research results the best effect in the meaning of processing approach that allows obtaining the most developed in terms of thickness and porosity HAp layer. The crystal size and morphological aspects of the HAp were not considered as also other topological issues. Further research is necessary, however, the beast correlation needs a cell's response.
Abstract-Authors should be more specific on what was the “best effect”.
Thank you for your suggestion. This general phrase has been replaced with a more specific definition. (see p.1 l.22-24)
Line 58- The term “bio-film” could remind most readers of the film created by bacteria. Authors need to rephrase the term.
We agree with the provided comment. The “Bio-film” phrase was removed from the article.
Figure 1 Please spell out MA, though it was spelled later in the text.
Thank you for this comment. We have corrected the figure and spelled the process name.
Figure 4 What is the difference between a and c? They looked like images with the same magnification.
The magnification is the same, however, the samples are after different processes. The samples in “Fig. 4. a” are after the preparation of metallographic specimens and etching in Kroll’s reagent. On these SEM microphotographs, we wanted to show the microstructure of base materials. On the other hand, the samples in “Fig. 4. c” are after the first step of surface treatment which is acid etching. These SEM micrographs present the surface reaction to acid etching.
We would like to thank you once again for your suggestions for improving our manuscript.
Yours faithfully,
A. Miklaszewski

Reviewer 2 Report
In the present work by Piechowiak et al. the authors investigate the formation of hydroxyapatite coatings on modified TiMo alloys. The influence of chemical composition of starting alloys and alkaline etching on the deposition of coatings is studied. This manuscript presents some interesting results, however, I do not recommend it for publication in the present form. In my opinion it could be reconsidered after a major revision is done.
According to experimental section the Ca to P ratio in the reaction medium during the hydrothermal treatment is 1 to 1. Why did the author choose such ratio, which is very different compared to the Ca/P ration in hydroxyapatite (1.67)?
How do the authors explain that substitution of Ti by smaller Mo atoms (lines221-223, Fig. 3) results in larger interplanar distances? This observation is in contradiction with the basics of X-ray diffraction analysis.
XRD patterns cannot be called "spectra"
How do the authors explain the differences in the amount of HAp phase on differently prepared substrates (Figs. 9-11). The intensity of HAp peaks on 12-HT substrates is very low compared to others.
The authors mention the importance of the coating morphology for practical applications. I would suggest to compare the obtained morphology with the morphologies of HAp coatings produced by other wet chemical methods.
Do the authors see any trends in the morphology of HAp coatings related to the substrate preparation? Fig. 12.
Did the authors investigate the adhesion of coatings?
Why did the authors took glycerol for wettability tests? This experiment must be re-done with water.
Conclusions are too long. Part of this text should be moved to the Results and discussion section.
English of the manuscript should be improved. Some sentences are hardly understandable.
Author Response
Dear Reviewer 2,
We would like to appreciate you allowing us to resubmit our manuscript: Low-temperature hydrothermal treatment surface functionalization of the ultrafine-grained TiMo alloys for medical applications to Materials. We are pleased with your comments and objective feedback considering the draft of our article. The changes were marked in our article in red colors.
All of the suggestions were included during the correction of our article and the comments were appropriately responded:
In the present work by Piechowiak et al. the authors investigate the formation of hydroxyapatite coatings on modified TiMo alloys. The influence of chemical composition of starting alloys and alkaline etching on the deposition of coatings is studied. This manuscript presents some interesting results, however, I do not recommend it for publication in the present form. In my opinion it could be reconsidered after a major revision is done.
According to experimental section the Ca to P ratio in the reaction medium during the hydrothermal treatment is 1 to 1. Why did the author choose such ratio, which is very different compared to the Ca/P ration in hydroxyapatite (1.67)?
The treatment procedure was selected based on our previous research on magnesium alloys, where the effectiveness in the creation of the HAp coating has been confirmed during hydrothermal treatment in the proposed medium. Furthermore, the previous choice of the reaction medium was made based on literature analysis see below:
- Andrzej Miklaszewski, Kamil Kowalski, Mieczyslaw Jurczyk, Hydrothermal Surface Treatment of Biodegradable Mg-Materials, Metals 2018, 8(11), 894; https://doi.org/10.3390/met8110894
- Masanari Tomozawa, Sachiko Hiromoto, Growth mechanism of hydroxyapatite-coatings formed on pure magnesium and corrosion behavior of the coated magnesium, Applied Surface Science Volume 257, Issue 19, 15 July 2011, Pages 8253-8257, https://doi.org/10.1016/j.apsusc.2011.04.087
- Sachiko Hiromoto, AkikoYamamoto, High corrosion resistance of magnesium coated with hydroxyapatite directly synthesized in an aqueous solution, Electrochimica Acta Volume 54, Issue 27, 30 November 2009, Pages 7085-7093, https://doi.org/10.1016/j.electacta.2009.07.033
- Masanari Tomozawa, Sachiko Hiromoto, Yoshitomo Harada, Microstructure of hydroxyapatite-coated magnesium prepared in aqueous solution, Surface and Coatings Technology Volume 204, Issue 20, 15 July 2010, Pages 3243-3247, https://doi.org/10.1016/j.surfcoat.2010.03.023
How do the authors explain that substitution of Ti by smaller Mo atoms (lines221-223, Fig. 3) results in larger interplanar distances? This observation is in contradiction with the basics of X-ray diffraction analysis.
We appreciate that comment, the proper explanation was added in the main manuscript text (see p.6 l. 221-226) for details of pointed publication see tab 3.
XRD patterns cannot be called "spectra"
Thank you for this comment. We have replaced the “XRD spectra” phrase with the “XRD patterns”
How do the authors explain the differences in the amount of HAp phase on differently prepared substrates (Figs. 9-11). The intensity of HAp peaks on 12-HT substrates is very low compared to others.
Thank you for this comment. a proper explanation were added to the main manuscript text (see p.16 l.347-352 and p.18 l.375-390)
The authors mention the importance of the coating morphology for practical applications. I would suggest to compare the obtained morphology with the morphologies of HAp coatings produced by other wet chemical methods.
Thank you for this comment. The importance of the coating morphology mentioned by the authors in a practical application concept, focuses on the further planned step recognition for the drug encapsulation and delivery systems.
Do the authors see any trends in the morphology of HAp coatings related to the substrate preparation? Fig. 12.
Thank you for this comment. The authors observed the relations in the morphology of HAp coatings related to the substrate preparation and mentioned in the manuscript's main text. (see p.9 l.262-271; p.13 l.305-310 and l.312-315; p.16 l.338-339;)
Did the authors investigate the adhesion of coatings?
Thank you for this comment, however, the adhesion tests were not performed at this stage of research. In the present reaserch autors try to define an influence of the molar concentration of NaOH (5M, 7M, 10M, 12M) during the alkaline treatment on the growth of hydroxyapatite crystals. Further tests remain awaited, and the layer adhesion stays an important factor in any applicable considerations which for this research make only an introduction.
Why did the authors took glycerol for wettability tests? This experiment must be re-done with water.
Thank you for this comment. At the initial stage of research consider water CA measurements were done, however, they didn't give a measurable result. Due to the high porosity of the coating, it was necessary to use a liquid with a high viscosity which allows registering the contact angle relation.
Conclusions are too long. Part of this text should be moved to the Results and discussion section.
Thank you for this significant comment. The conclusions have been shortened and specified.
English of the manuscript should be improved. Some sentences are hardly understandable.
The authors agree with that comment. The article was proofread by a native speaker and corrected to be understandable in all sentences.
We would like to thank you once again for your suggestions for improving our manuscript.
Yours faithfully,
A.Miklaszewski

Reviewer 3 Report
Dear Authors,
Regarding your manuscript, I have only two comments/suggestions to do:
- For Figure 1, along with the XRD analysis, I suggest that you also indicate the SEM analysis and the determination of wettability, as these analyzes have been done, are important for experimental program and should also be mentioned.
- At the end of #3.1 part I suggest to indicate the average of the grain dimension obtained for the studied alloys, in order to prove the ultrafine character of the obtained micro-structure, not only by images but also by specific values.
Author Response
Dear Reviewer 3,
We would like to appreciate you allowing us to resubmit our manuscript: Low-temperature hydrothermal treatment surface functionalization of the ultrafine-grained TiMo alloys for medical applications to Materials. We are pleased with your comments and objective feedback considering the draft of our article. The changes were marked in our article in red colors.
All of the suggestions were included during the correction of our article and the comments were appropriately responded:
For Figure 1, along with the XRD analysis, I suggest that you also indicate the SEM analysis and the determination of wettability, as these analyzes have been done, are important for the experimental program and should also be mentioned.
Thank you for this comment. We have corrected figure 1 and added the descriptions of suggested processes in proper places.
At the end of #3.1 part I suggest to indicate the average of the grain dimension obtained for the studied alloys, in order to prove the ultrafine character of the obtained micro-structure, not only by images but also by specific values.
Thank you for your suggestion. We have added a new Fig. 4. with recommended analysis. An additional comment at this place remains needed. An ultrafine-grained (UFG) materials define with grain sizes in the range from 10 to 1000 nm, next fine-grained (FG) group opens the range from 2µm. The obtained results confirm the UFG material character.
We would like to thank you once again for your suggestions for improving our manuscript.
Yours faithfully,
A.Miklaszewski

Round 2
Reviewer 1 Report
Please check if reference #31 was correct.
All other issues I raised were addressed appropriately.
Author Response
We would like to appreciate you allowing us to resubmit our manuscript: Low-temperature hydrothermal treatment surface functionalization of the ultrafine-grained TiMo alloys for medical applications to Materials. We are pleased with your comments and objective feedback considering the draft of our article. The changes were marked in our article in red colour.
All of the suggestions were included during the correction of our article and the comments were appropriately responded:
The reference #31 was correct.
Reviewer 2 Report
accept
Author Response
We would like to appreciate you allowing us to resubmit our manuscript: Low-temperature hydrothermal treatment surface functionalization of the ultrafine-grained TiMo alloys for medical applications to Materials. We are pleased with your comments and objective feedback considering the draft of our article. The changes were marked in our article in red colours.
All of the suggestions were included during the correction of our article and the comments were appropriately responded.
A language check was made.